# Folate Deficiency Enhanced Inflammation and Exacerbated Renal Fibrosis in High-Fat High-Fructose Diet-Fed Mice

**DOI:** 10.3390/nu15163616

**Published:** 2023-08-17

**Authors:** Chun-Wai Chan, Bi-Fong Lin

**Affiliations:** Department of Biochemical Science and Technology, College of Life Science, National Taiwan University, Taipei 10617, Taiwan; chunwai@ntu.edu.tw

**Keywords:** folate deficiency, leptin, chronic inflammation, fibrosis, obesity-related nephropathy

## Abstract

The prevalence of obesity and chronic kidney disease (CKD) is increasing simultaneously and rapidly worldwide. Our previous study showed that folate deficiency increased lipid accumulation and leptin production of adipocytes. Whether folate plays a role in CKD, particularly obesity-related nephropathy remains unclear. To investigate the effects of folate deficiency on CKD in diet-induced obese mice, four groups of male C57BL/6 mice were fed either a normal-fat diet (NF) with folate (NF+f); NF without folate (NF−f); high-fat high-fructose diet (HFF) with folate (HFF+f); or HFF without folate (HFF−f) for 12 months during the study. The results showed that HFF increased not only body weight, fasting blood glucose, total cholesterol (TC), low-density lipoprotein (LDL)-cholesterol, and blood pressure, but also cytokines levels, such as interleukin (IL)-2, interferon (IFN)-γ, IL-17A/F, IL-6, monocyte chemoattractant protein (MCP)-1, and transforming growth factor (TGF)-β1. The indicators of kidney failure including urinary protein, neutrophil gelatinase-associated lipocalin (NGAL), renal type I and IV collagen deposits and leptin content, and serum creatinine were also increased by HFF. Folate-deficient diets further elevated serum TC, LDL-cholesterol, IL-6, tumor necrosis factor (TNF)-α, MCP-1, TGF-β1, and leptin, but decreased IL-10 level, and thus exacerbated renal fibrosis. To investigate the possible mechanisms of folate deficiency on renal injury, phosphorylation of pro-fibrosis signaling molecules, including signal transducer and activator of transcription (STAT)3 and small mothers against decapentaplegic (Smad)2/3, were assayed. Both HFF and folate deficiency significantly increased the phosphorylation of STAT3 and Smad2/3, suggesting synergistic effects of HFF−f on chronic renal inflammation and fibrosis. In conclusion, the results demonstrated that folate deficiency might aggravate inflammatory status and enhance renal fibrosis.

## 1. Introduction

Overweight and obesity are growing global public health problems. Hypertrophic adipocytes tend to increase leptin and pro-inflammatory cytokines levels, immune cell infiltration, hypoxia, and insulin resistance [1]. Excessive body fat accumulation is a major cause of hyperlipidemia, type 2 diabetes, and hypertension; these comorbidities are the risk factors for the development of chronic kidney disease (CKD) [2]. In 2017, CKD caused 1.2 million deaths worldwide, as well as 7.6% of all cardiovascular disease deaths related to impaired kidney function [3]. The obesity epidemic has led to an increased incidence of obesity-related nephropathy in American, European, and Asian cohorts over the past three decades [4].

Pathological features of obesity-related nephropathy include proteinuria, glomerulomegaly, glomerulosclerosis, and renal functional deterioration [5]. The glomerulus expands in response to obesity-induced high renal blood flow, glomerular filtration fraction, filtration rate, and glomerular pressure [6]. The measurement of urine protein, serum creatinine, blood urea nitrogen (BUN), and estimated glomerular filtration rate are the clinical hallmarks used to assess renal function and CKD progression. Recently, a 25-kDa protein known as neutrophil gelatinase-associated lipocalin (NGAL) released in blood and urine upon kidney injury is regarded to be a primary indicator of renal damage. Elevated urinary NGAL level is a sensitive marker to reflect the severity of renal impairment in subjects affected by CKD, especially in glomerular disease [7].

Modern diets with high-fat foods, high-fructose corn syrup-sweetened beverages, and fewer vegetables might result in inadequate folate intake. Folate is a water-soluble B-group vitamin essential for DNA synthesis and cell division. A recent study found that a lower dietary intake of folate was associated with an increased risk of CKD stage 3B and over [8]. Obesity and unhealthy diet have been reported to be associated with poor folate status [9]. Our previous study showed that folate deficiency enhanced lipid accumulation and leptin production of adipocytes [10]. Circulating leptin is actually elevated in obese people due to leptin resistance, and regarded as a uremic toxin, frequently exacerbates the progression of obesity-related nephropathy [11,12,13]. In addition, leptin further stimulates the production of pro-inflammatory mediators such as monocyte chemotactic protein (MCP)-1, tumor necrosis factor (TNF)-α, and interleukin (IL)-6 by macrophages [14]. Folate deficiency also increased IL-6, TNF-α, and MCP-1 secretions of RAW 264.7 macrophage cell line [15], and enhanced intestinal inflammation due to decreased colonic regulatory T (Treg) cells in mice fed a folate-deficient diet [16]. Our previous study further demonstrated that folate deficiency inhibited the differentiation of naïve T cells to Treg cells and also the secretion of anti-inflammatory IL-10 cytokine [17]. Therefore, chronic higher leptin secretion by expanded adipocytes might lead to persistent systemic inflammation. The interaction between adipose tissue-derived adipokines and the kidneys, referred to as the adipose–renal axis, has been proposed to be important for pathological changes in the kidneys [18,19]. Therefore, whether folate deficiency affects renal function is worthy of study.

Renal function is critical for homeostasis in vivo. In diabetic or high-calorie diet-induced metabolic disruption mice, excess lipid and cholesterol ester accumulations occur in the kidney via a sterol regulatory element-binding protein-1c-dependent pathway [20], leading to lipid toxicity and oxidative stress to the renal cells and activating the pro-inflammatory pathways [21,22]. Renal resident cells such as podocytes, mesangial cells, tubular epithelial cells, and endothelial cells secrete the pro-inflammatory MCP-1, IL-6, and pro-fibrotic transforming growth factor (TGF)-β1 during the transition from normal to pathological changes in the kidneys [23,24,25]. MCP-1 is a chemokine that attracts immune cells to the inflamed kidney, and this recruitment leads to higher production of MCP-1, IL-6, and TNF-α, subsequently causing renal injury and dysfunction. Persistent renal inflammation promotes the progression of renal fibrosis, a final common outcome of CKD. The development of novel anti-fibrotic therapies to prevent the progression of CKD has been a focus of research. TGF-β1 plays a critical stimulus for kidney fibrosis while IL-10 acts as an anti-fibrotic cytokine in kidney disease [26]. Therefore, the effects of folate deficiency on renal inflammation and fibrosis were investigated in high-fat high-fructose diet-induced obese mice in this study.

To understand the mechanisms of the effect of diets on renal fibrosis, we further investigated the possible signal pathways involved. Several reviews summarized the response to leptin, IL-6, or high glucose, the signal transducer and activator of transcription (STAT)3 can be activated by tyrosine phosphorylation at tyrosine residues via Janus tyrosine kinases in renal cells, which might contribute to glomerulosclerosis and renal fibrosis [18,24,27]. Phosphorylated STAT3 induces the expression of fibrotic mediators, such as TGF-β1, type I, and type IV collagen [28]. Diet-induced obese rats were found to have increased STAT3 signaling and renal inflammation [29], suggesting the association between STAT3 and CKD. Moreover, STAT3 also contributes to the expression of TGF-β1 that further binds to its receptors and activates small mothers against decapentaplegic (Smad)2/3. It is well-known that Smad2/3 is extensively activated in the fibrotic kidney in both patients and animal models with CKD. The TGF-β1/Smad axis regulates the transcription of target genes similar to the STAT3 pathway to induce fibrotic outcomes [25]. Therefore, whether folate deficiency might enhance renal fibrosis via activation of STAT3 and Smad2/3 signaling is worthy of investigation.

The aim of the present study is to investigate the effects of dietary folate on renal inflammation and fibrosis in diet-induced obese mice. Our data demonstrated that folate deficiency might exacerbate obesity-related nephropathy in mice by enhancing inflammation.

## 2. Materials and Methods

### 2.1. Experiment Animals and Composition of Diets

The six-week-old male C57BL/6 mice (n = 54) were purchased from National Laboratory Animal Center (Taipei, Taiwan, ROC), and single-housed in cages with ad libitum access to AIN-93 diet and water, maintained on a 12 h light/12 h dark cycle under controlled environment at 23 ± 2 °C and 55 ± 5% relative humidity. After 2 weeks of acclimatization, mice were assigned to either a normal-fat diet (NF) with folate (NF+f, n = 11), NF without folate (NF−f, n = 11), high-fat high-fructose diet (HFF) with folate (HFF+f, n = 16), or HFF without folate (HFF−f, n = 16). Composition of NF (7% soybean oil, 52.95% corn starch, 10% sucrose, 20% casein, 0.3% l-cysteine, 5% cellulose, 3.5% mineral mix, 1% vitamin mix without folate, and 0.25% choline bitartrate) and HFF (22.6% lard, 1% soybean oil, 10% sucrose, 31.6% fructose, 23% casein, 0.4% l-cysteine, 5.7% cellulose, 4.2% mineral mix, 1.2% vitamin mix without folate, and 0.3% choline bitartrate) were adjusted to maintain the similar daily intake of essential nutrients except folate. Folic acid was added back to NF+f and HFF+f (2 mg/kg diet), and NF−f diet (0.2 mg/kg diet) [10]. These 2-month-old mice were fed for 12 months until the day of sacrifice. The body weight and feed intake of each mouse were recorded once a week.

### 2.2. Measurement of Serum Folate Level and Biomarkers

Blood samples were collected in a laminar airflow chamber from mice at 0, 1, and 12 months after test diet feeding to monitor folate status changes, then centrifuged at 12,000 rpm to obtain sterile serum, and samples were stored at −80 °C until analysis. Folate concentration in serum was measured by *Lactobacillus casei* (ATCC 7469) microbiological assay as previously described [10]. Sera collected at 12 months after the test diet was assayed for biomarkers. Serum homocysteine (USCN, Wuhan, China), IL-6 and MCP-1 (BioLegend, San Diego, CA, USA), and TGF-β1 (Invitrogen, Carlsbad, CA, USA) levels were detected using an enzyme-linked immunosorbent assay (ELISA) kit. Serum creatinine levels were measured using the QuantiChrom™ creatinine assay kit (BioAssay Systems, Hayward, CA, USA), following the manufacturer’s protocol.

### 2.3. Measurement of Fasting Blood Glucose and Serum Cholesterol

Overnight fasting blood samples were collected at 0 and 12 months after test diet feeding. The fasting blood glucose levels were detected using a Contour Glucose Meter (Bayer, Leverkusen, NRW, Germany) and Contour Glucose Strips (Bayer). The fasting serum total cholesterol (TC) and low-density lipoprotein cholesterol (LDL-C) were analyzed using quantification enzymatic kits (Randox, Antrim, UK), according to the manufacturer’s instructions.

### 2.4. Measurement of Blood Pressure and Heart Rate

Before sacrifice, the systolic blood pressure, diastolic blood pressure, and heart rate of each mouse were measured in the conscious state using a small animal tail-cuff blood pressure analyzer (MK2000; Muromachi, Tokyo, Japan). Three consecutive measurements were performed to obtain the mean blood pressure value for each mouse.

### 2.5. Determination of Renal Function Biomarkers

Urine samples from mice were collected at 0, 6, 8, 10, and 12 months after test diet feeding to monitor renal function. Urine protein concentrations were measured using Coomassie protein assay reagent (ThermoFisher Scientific, Rockford, IL, USA) and normalized by urine creatinine (BioAssay Systems, Hayward, CA, USA). Urinary neutrophil gelatinase-associated lipocalin (NAGL) content was determined using an ELISA kit (R&D Systems, Minneapolis, MN, USA), following the manufacturer’s protocol.

### 2.6. Assay of Cytokine Secretions from Primary Splenocytes

After 12 months of test diet feeding, the mice were sacrificed and single-cell suspensions of splenocytes isolated from spleens of mice were collected as previously described [30]. Splenocytes (5 × 10^6^ cells/mL) from NF+f, HFF+f, NF−f, or HFF−f diet-fed mice were cultured in corresponding folate-containing (2.3 μM folic acid) or folate-free RPMI 1640 medium (Gibco, Grand Island, NY, USA) with TCM serum replacement (Protide Pharmaceuticals, Crystal Lake, IL, USA) and 1% antibiotic. To determine cytokine secretion, splenocytes were activated with concanavalin A (ConA; 2 μg/mL, T cell mitogen, Sigma-Aldrich, St. Louis, MO, USA) or LPS (10 μg/mL, B cell mitogen, Sigma-Aldrich, St. Louis, MI, USA) for 48 h. The cytokine levels, including IL-2, interferon (IFN)-γ, IL-17A/F, TNF-α, IL-6, IL-4, and IL-10, in the supernatant were analyzed by ELISA (BioLegend), according to the manufacturer’s instructions.

### 2.7. Histopathological Analysis

Kidney tissues were harvested, removed the renal fascia, fixed with 10% neutral buffered formalin (Sigma), and then embedded in paraffin. The 4 μm thickness tissue sections were stained with periodic acid Schiff (PAS), and Masson’s trichrome (Masson) after deparaffinization and rehydration. Images were captured on an IM-3 LED light microscope (Optika, Bergamo, Italy) equipped with an 8.3-megapixel digital camera (Optika) using the ProView software (Optika). Then, the glomerulus size and renal fibrotic area were quantified using ImageJ software version 1.52a.

### 2.8. Determination of Cytokines and Collagen Contents in Renal Tissue Homogenates

To determine the level of cytokines, kidney tissues were harvested, weighed, and rinsed with ice-cold phosphate-buffered saline. Then, tissues were homogenized using radio immune precipitation (RIPA) lysis buffer (BioBasic, Markham, ON, Canada) with a halt protease inhibitor cocktail (ThermoFisher, Waltham, MA, USA). Homogenates were centrifuged at 12,000× *g* for 10 min to collect the supernatant for analysis of biomarkers, including leptin (R&D Systems, Minneapolis, MN, USA), TGF-β1 (Invitrogen), MCP-1, IL-6, TNF-α, and IL-10 (BioLegend), type I and IV collagen (USCN) by ELISA commercial kit. The values were normalized to protein contents, as determined with bicinchoninic acid (BCA) assay (ThermoFisher, Waltham, MA, USA).

### 2.9. Western Blot Analysis

Kidney tissues were homogenized using RIPA lysis buffer (BioBasic) containing protease inhibitor cocktail (ThermoFisher, Waltham, MA, USA) and PhosSTOP™ phosphatase inhibitor (Roche, Basel, Switzerland). Homogenates were centrifuged at 20,000× *g* for 30 min to collect the supernatant and protein contents were measured by BCA assay (ThermoFisher, Waltham, MA, USA). Tris-glycine-sodium dodecyl sulfate-polyacrylamide gel electrophoresis was conducted to separate the equal amounts of protein lysate. Then, proteins were electroblotted onto polyvinylidene difluoride membranes (Roche), blocked, and incubated with rabbit or mouse monoclonal antibodies against STAT3 (Proteintech, Rosemont, IL, USA), phospho-STAT3, Smad2/3, phospho-Smad2/3 (Cell Signaling Technology, Danvers, MA, USA), and GAPDH (Abcam, Cambridge, UK) at 4 °C for 12 h, rinsed and incubated with HRP-conjugated secondary antibodies (Jackson ImmunoResearch, West Grove, PA, USA) at 4 °C for 1 h. The Western blot image was visualized by the HRP-substrate and UVP Imaging System, quantified using the VisionWorks software version 8.21, and normalized to GAPDH.

### 2.10. Statistical Analysis

All data were presented as mean ± standard error of the mean (SEM). Statistical analyses were performed in SPSS software version 22.0 (IBM Corp., Armonk, NY, USA) and Prism 7.0 software (GraphPad, La Jolla, CA, USA). All data sets were tested for normality using the Shapiro–Wilk test and the *p* values of data sets were greater than 0.05, indicating that data sets were normally distributed. Data were analyzed using two-way analysis of variance (ANOVA) to determine the main effects (folate or/and HFF diet) and interaction (folate × HFF), followed by Duncan’s post hoc test as previously described [31]. Also, the Pearson correlation test was used. *p* < 0.05 or different letters was considered statistically significant.

## 3. Results

### 3.1. The Highest Body Weight, Serum Cholesterol, and Systolic Blood Pressure in Mice Fed High-Fat High-Fructose Diet without Folate

To investigate whether folate deficiency increased lipid accumulation and also caused metabolic disorder in mice fed a high-fat high-fructose (HFF) diet, we fed mice either normal-fat (NF) or HFF diet with folate (NF+f, HFF+f) or without folate (NF−f, HFF−f). There were no significant differences in body weight, total cholesterol (TC), low-density lipoprotein cholesterol (LDL-C), blood glucose, sera folate, and homocysteine levels among the groups before diet treatment (Appendix A). After 12 months of feeding, HFF-fed mice had significantly higher body weight, fasting serum TC, LDL-C, blood glucose, and systolic blood pressure (Figure 1A–E). The trend for higher diastolic blood pressure in HFF mice did not reach statistical significance (*p* = 0.107) (Figure 1F).

Mice fed with folate-deficient NF−f or HFF−f had significantly lower serum folate and higher homocysteine as expected (Figure 1G,H). Folate deficiency significantly increased body weight, fasting TC, and LDL-C (Figure 1A–C). Therefore, mice fed HFF−f had the highest body weight, serum cholesterol levels, and systolic blood pressure. The results were in accordance with the previous report that folate deficiency increased lipid accumulation. Systolic blood pressure of mice from four different diet groups was positively correlated with body weight (Figure 1I), revealing that inadequate folate status might also increase blood pressure, which is one of the risk factors for developing chronic kidney disease.

### 3.2. Higher Inflammatory Cytokine Productions by Splenocytes from Mice with Folate Deficiency

To investigate the impact of folate deficiency and HFF diet on systemic immune responses, the cytokines secreted by mitogen-activated splenocytes isolated from mice fed with different diets were determined. After 12 months of dietary intervention, the HFF diet not only significantly increased the spleen weight of mice (Figure 2A), but also promoted IL-2, IFN-γ, and IL-17A/F secretions of ConA-stimulated T cells (Figure 2B–D). However, folate status did not significantly affect spleen weight and also IL-2, IFN-γ, and IL-17A/F levels.

In contrast, folate deficiency significantly increased pro-inflammatory cytokine IL-6 and TNF-α secretions which were not significantly affected by HFF (Figure 2E,F). Neither folate nor HFF had significant effects on splenic IL-4 secretion analyzed by two-way ANOVA (Figure 2G). However, folate deficiency not only promoted pro-inflammatory but also suppressed anti-inflammatory cytokine IL-10 production by both T cells and B cells, represented by ConA- or LPS-stimulated splenocytes, respectively (Figure 2H,I). The results indicated that folate deficiency might enhance pro-inflammatory immune responses which has been considered as an important component of chronic kidney disease.

### 3.3. Higher Serum and Urinary Biomarkers of Kidney Dysfunction in Mice with Folate Deficiency

To explore whether kidney function was affected by chronic inflammation driven by folate deficiency and HFF diets, representative serum and urine indicators were monitored during the feeding periods. After 12 months of dietary intervention, HFF diets significantly increased serum levels of IL-6, MCP-1, TGF-β1, and creatinine (Figure 3A–D), and urinary protein and neutrophil gelatinase-associated lipocalin (NGAL) (Figure 3E,F). Although folate deficiency did not show significant effects on these serum markers due to folate × HFF interaction, the HFF−f mice had the highest serum levels of chemokine MCP-1, and the pro-fibrotic marker TGF-β1 among the dietary groups.

In addition, folate deficiency also significantly increased urinary protein and NGAL excretion (Figure 3E,F), suggesting that renal function might be further worsened by folate deficiency. Since TGF-β1 plays a critical stimulus for kidney fibrosis during the transition from normal to pathological changes in the kidney, serum MCP-1 level (Figure 3G), serum creatinine level (Figure 3H), and urine protein level (Figure 3I) were found to be significantly correlated with serum TGF-β1 levels. These results suggested that both folate deficiency and HFF might enhance kidney injury and dysfunction in mice.

### 3.4. Higher Perirenal White Adipose Tissue, Renal Leptin, and Renal TGF-β1 Contents in Mice with Folate Deficiency

The impact of folate deficiency and the HFF diet on the kidneys was examined by lipid accumulation and pro-fibrotic mediator contents. HFF significantly increased kidney weight, but not folate (Figure 4A). However, both HFF diet and folate deficiency significantly promoted perirenal white adipose tissue (WAT) mass (Figure 4B), and thus renal leptin contents (Figure 4C). Interestingly, renal TGF-β1 levels were also increased by the HFF diet and folate deficiency (Figure 4D). The perirenal WAT, renal leptin, and TGF-β1 in HFF−f mice were the highest among the groups.

Renal leptin levels were positively correlated to perirenal WAT mass (Figure 4E), suggesting that leptin from renal fat tissue might contribute to renal inflammation. In addition, there was a positive correlation between renal contents of TGF-β1 and leptin (Figure 4F). Our result indicated that both the HFF diet and folate deficiency promoted the development of renal inflammation and fibrosis.

### 3.5. Higher Renal Pro-Inflammatory but Lower Anti-Inflammatory Cytokines in Mice with Folate Deficiency

The levels of the pro-inflammatory cytokines such as MCP-1, IL-6, and TNF-α, as well as anti-inflammatory cytokine IL-10, in kidney homogenates were further determined to evaluate renal inflammation. Both HFF diet and folate deficiency increased renal MCP-1 and IL-6 contents (Figure 5A,B), as well as TNF-α. The highest renal TNF-α contents were detected in the HFF−f mice, despite folate × HFF interaction (Figure 5C). The highest macrophage marker *F4/80* expression in the kidneys of the HFF−f mice suggested that folate deficiency and the HFF diet increased the inflammatory cell infiltration in the kidney (Appendix A). In contrast, both HFF diet and folate deficiency decreased anti-inflammatory cytokine IL-10 contents in the kidney (Figure 5D). The NF−f mice had the lowest renal IL-10 among the dietary groups due to an interaction between the effects of folate deficiency and HFF.

As shown in Figure 5E, there was a positive correlation between renal contents of MCP-1 and leptin, implying that renal leptin, might play a role in local inflammation. In addition, there was a negative correlation between IL-6 and IL-10 (Figure 5F), suggesting that renal anti-inflammatory cytokine might be important in the regulation of renal inflammation.

### 3.6. Exacerbated Renal Extracellular Matrix Depositions and Fibrosis in Mice with Folate Deficiency

Since both folate deficiency and the HFF diet enhanced pro-fibrotic and inflammatory cytokines in the kidneys of mice, renal histopathology and collagen contents were examined. Periodic acid Schiff-stained kidney sections showed a larger glomerulus size (glomerulomegaly) in both HFF-fed mice and Masson-stained sections showed more fibrotic lesions around the glomerulus (glomerulosclerosis) in HFF−f mice (Figure 6A). Image quantification revealed that folate deficiency also significantly worsened glomerulomegaly and glomerulosclerosis. The HFF−f mice had the largest glomerulus size and renal fibrosis area percentage among the dietary groups (Figure 6B,C).

In addition, the most common collagen components in renal fibrosis were assayed. Both folate deficiency and HFF diet significantly elevated renal type I collagen, a biomarker for interstitial fibrosis (Figure 6D). The main component of glomerular basement membrane type IV collagen in the kidneys was also increased by folate deficiency and the HFF diet (Figure 6E). These results indicated that folate deficiency further exacerbated renal extracellular matrix depositions and fibrosis in mice fed with an HFF diet.

### 3.7. Folate Deficiency Promoted Renal Pro-Fibrosis Signaling Activation in Mice

To confirm the effects of folate deficiency on renal fibrosis, the activation by phosphorylation of pro-fibrosis signaling molecules, STAT3 and Smad2/3 were assayed by Western blotting (Figure 7A). Image quantification revealed that both folate deficiency and HFF significantly promoted phosphorylated STAT3 and phosphorylated Smad2/3 in mice (Figure 7B,C). As a result, the signals of STAT3 and Smad2/3 phosphorylation were the highest in the kidneys of mice fed with an HFF−f diet.

In addition, renal leptin level was found to be positively correlated with STAT3 signaling activation (Figure 7D), confirming the association between leptin and STAT3 activation. Renal TGF-β1 level was also correlated with Smad2/3 signaling activation (Figure 7E), confirming the association between TGF-β1 and Smad2/3 activation. These results obtained directly from the kidneys of mice long-term fed the HFF diet without folate showed significantly increased STAT3 and Smad2/3 activation that contribute to the development of kidney fibrosis. The synergistic effects of folate deficiency and HFF on chronic renal inflammation and fibrosis were worthy of notice.

## 4. Discussion

Our results showed that long-term dietary folate deficiency exacerbated renal fibrosis in mice by enhancing inflammation focused on the kidneys. There are only very few studies on folate and CKD published. From the references, the study on whether folate levels were changed in obese patients with nephropathy has not been found so far. However, some references mentioned that serum folate levels negatively correlated with the estimated glomerular filtration rate in patients with CKD, the lower dietary intake of folate was associated with an increased risk of CKD stage 3B and over [8,32]. Folate participates in the regeneration of methionine from homocysteine, and thus folate deficiency might result in hyperhomocysteinemia. Observational studies have shown that higher plasma homocysteine is a risk factor for developing CKD [33,34,35]. Hypertension is also an important risk factor for kidney injury, which exerts a high fluid shear stress on renal resident cells, further prompting renal cell hypertrophy and detachment, and TGF-β1 production [4]. The current study showed that HFF−f mice had the highest systolic blood pressure among all the groups. The systolic blood pressure was positively correlated with body weight that increased by HFF diet and folate deficiency, which might be one of the reasons that obese people had a higher risk of CKD and renal dysfunction compared to those of lean individuals [36,37].

There are few studies describing the effect of folate status on blood glucose and cholesterol. One study mentioned that hyperglycemia and higher very-low-density lipoprotein-cholesterol were noted in female Institute of Cancer Research (ICR) mice fed a folate-deficient diet for 24 weeks [38]. It is well-known that homocysteine is increased by poor folate status. Although one study showed that insulin signaling was impaired by homocysteine thiolactone in hepatoma cell line [39], the increase in fasting blood glucose by folate deficiency in our study did not reach a significant difference yet. The link between homocysteine and cholesterol was found to increase the 3-hydroxy-3-methylglutaryl coenzyme A reductase for cholesterol synthesis by homocysteine [40]. Our study demonstrated that folate deficiency not only increased serum homocysteine but also fasting TC and LDL-C in mice. Although serum cholesterol might vary in CKD patients, there are few studies investigating the association between cholesterol and the development of CKD. We found significantly positive correlations between serum cholesterol level and renal collagen deposit, glomerulus size, renal fibrosis area, serum creatinine, urinary protein, and urinary NGAL (Appendix A). Our data suggested that elevated serum cholesterol by either folate deficiency or HFF might contribute to the progress and development of CKD.

The impacts of folate deficiency and the HFF diet on systemic immune responses were evaluated by the functions of splenocytes. Our results showed that C57BL/6 mice fed the HFF diet had significantly larger spleen compared to those fed a normal-fat diet, which was similar to another study [41]. The spleen is the largest peripheral lymphoid organ in the body, containing a large number of immune cells such as T and B cells that secret cytokines and modulate the systemic immune response. Our study demonstrated that T-cell cytokines, such as IL-2, IFN-γ, and IL-17A/F were significantly increased by the HFF diet, but not folate deficiency. An in vitro study reported that folate deficiency increased IL-6, TNF-α, and MCP-1 secretions of macrophage cell lines [15]. In our in vivo study, folate deficiency also increased pro-inflammatory cytokines TNF-α and IL-6, but not IL-4, by activated T cells, suggesting that folate deficiency might drive T-cell responses towards a pro-inflammatory phenotype. Higher anti-inflammatory cytokine IL-10 secretion of ConA-stimulated splenocytes of HFF groups might be explained by the compensation for low-grade systemic inflammation caused by a high-fat diet [42]. Folate-deficient mice had significantly lower secretions of IL-10 by not only ConA-stimulated T cells but also LPS-stimulated B cells, which is in accordance with our previous report that significantly lower IL-10 secreted by ConA-stimulated splenocytes from BALB/c mice fed with the folate-deficient diet for 13 weeks [17]. The importance of spleen-derived IL-10 in protection against obesity-related CKD has been demonstrated by the study with IL-10 knockout mice [43]. It is suggested that the down-regulatory effect of IL-10 on obesity-induced chronic systemic inflammation might play a role in terms of nephroprotection.

For serum biomarkers, cytokine concentrations are normally detected at low levels in healthy subjects but increased in obese subjects [44]. In our study, mice fed with the HFF diet had significantly higher body weight also with higher sera levels of IL-6 and MCP-1, which was in accordance with the observation that IL-6 and MCP-1 exerted a pro-inflammatory activity in obese-mediated inflammation [45]. Serum indicators were not significantly affected by folate deficiency. However, urinary markers for renal function, such as urine protein and NGAL, were significantly affected not only by the HFF diet but also by folate deficiency, suggesting that both the HFF diet and folate insufficiency could be the risk factors for CKD. We have previously shown that both HFF diet and folate deficiency increased lipid accumulation and leptin production [10]. In the present study, we further demonstrated that obesity-induced inflammation might be an underlying mechanism. In addition to the increase in pro-inflammatory cytokines secreted by splenocytes, we found that renal MCP-1, IL-6, and TNF-α were significantly higher but regulatory cytokine IL-10 was lower in mice fed HFF or folate-deficient diet after long-term feeding. Renal IL-10 was also negatively correlated with renal leptin content (r = −0.376, *p* = 0.005), agreeing with the hypothesis that leptin is a potential uremic toxin owing to its adverse effect on inflammation associated with CKD [12]. The positive correlation between renal levels of MCP-1 and leptin, and the negative correlation between IL-6 and IL-10 indicated that inflammation was related to obesity-related nephropathy. Therefore, HFF diet-induced obesity promoted chronic inflammation, and then poor dietary folate status might exacerbate renal dysfunction and inflammation in mice by enhancing systemic inflammation.

Our previous report showed that serum leptin significantly correlated not only with body weight, but also with perirenal white adipose tissue in mice fed with NF or HFF diet with or without folate [10]. In the current study, we also confirmed that both HFF diet and folate deficiency significantly increased renal leptin levels in mice. Leptin can freely pass through the glomerulus and then be reabsorbed by tubular cells due to its relatively small molecular size. The pathophysiological and clinical roles of leptin in CKD have been well-reviewed by Alix et al. [12]. The inability of the kidneys to clear circulating leptin might be the reason for elevated serum leptin levels in patients with moderate or advanced CKD [46]. However, the contribution of hyperleptinemia to the pathogenesis of CKD still needs to be clarified. The serum level of MCP-1 was positively correlated with serum leptin level (r = 0.416, *p* = 0.002) in our study. The renal pro-inflammatory cytokines MCP-1 significantly correlated with not only renal leptin content, but also serum leptin level (r = 0.628, *p* < 0.001). These positive correlations suggest that leptin plays a role in renal inflammation initiated by injured tubular epithelial cells to secrete MCP-1 for the recruitment of immune cells [23]. Leptin binds to a specific receptor in kidney resident cells such as mesangial cells, tubular cells, and podocytes, and then promotes TGF-β1 productions [47]. Further, STAT3 activation has been considered one of the signaling pathways mediated by leptin contributing to the pathological process in kidney fibrosis via an increment of TGF-β1 production and type I and IV collagen fiber depositions [18]. Inhibition of STAT3 in tubular epithelial cells prevented renal fibrosis in diabetic mice [28]. The results also showed that mice fed with high-fat diets had higher renal TNF-α and MCP-1 gene expression and urinary albumin and NGAL excretions [48]. It is suggested that dietary folate deficiency further aggravated renal inflammation and suppressed anti-inflammatory cytokine IL-10 secretion. Our previous study on folate deficiency increased leptin production of adipocytes implying that there are interactions among folate status, leptin, and immunoregulation.

TGF-β1 is a potent anti-inflammatory cytokine in response to renal injury to facilitate the repair process. However, TGF-β1 is also a key driver of renal fibrosis in CKD, suggesting its diverse role in kidney disease [49]. Nevertheless, it is well documented that TGF-β1 differently regulates renal inflammation and fibrosis through activating its downstream signal pathways such as phosphorylation of Smads [25]. In the present study, serum TGF-β1 was significantly correlated with serum MCP-1, creatinine, and urine protein, suggesting the association between inflammation and renal dysfunction with increased TGF-β1. Serum TGF-β1 might be a biomarker for chronic nephropathy progression [50]. Further, our result showed that renal TGF-β1 positively correlated with renal leptin, which is consistent with the mechanism that STAT3 signaling activation by leptin receptor increases renal TGF-β1 production of kidney resident cells during the development of fibrosis [18]. As a result, both HFF diet and folate deficiency promote chronic nephropathy progression so that HFF−f mice had the largest glomerulus size, fibrosis area, and highest renal collagen levels. In addition, the activation of STAT3 signaling and Smad2/3 signaling were examined directly using kidney tissue isolated from mice after 12 months of feeding in the present study. Activation of these pathways by phosphorylated STAT3 and Smad2/3 were affected by both HFF diet and folate deficiency. The positive correlation between leptin and phosphorylated STAT3 ratio, as well as TGF-β1 and phosphorylated Smad2/3 ratio in renal tissues, revealed the mechanisms of effects of the HFF diet and folate deficiency. In addition to STAT3 and Smad2/3 signaling, there are several crucial pathways involved in renal extracellular matrix deposition and fibrosis, such as the RAS superfamily, mammalian target of rapamycin complex 1, and hypoxia-inducible factor-1α [51,52]. These may be clues and strategies for us to further explore the mechanism in greater depth of folate nutritional status on renal fibrosis in the future.

## 5. Conclusions

In conclusion, our study demonstrated that dietary folate deficiency exacerbated high-fat high-fructose diet-induced inflammation and subsequent renal fibrosis in mice (Figure 8). Poor folate status might be unfavorable for immune regulation and play a critical role in facilitating the development of a pro-inflammatory microenvironment in the kidneys. Therefore, folate status might not be neglected in terms of chronic kidney disease in an aging society. It should be emphasized more that adequate folate intake is extremely important to maintain immune regulation and renal function.

## Figures and Tables

**Figure 1 nutrients-15-03616-f001:**
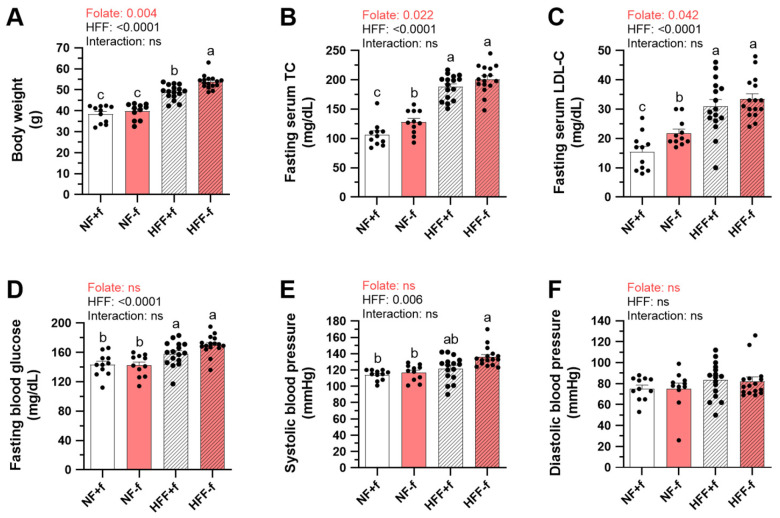
Folate deficiency increased body weight, fasting cholesterol, and systolic blood pressure of mice fed test diet for 12 months. (**A**) Body weight; (**B**) serum total cholesterol (TC); (**C**) serum low-density lipoprotein cholesterol (LDL-C); (**D**) blood glucose; (**E**) systolic blood pressure; (**F**) diastolic blood pressure; serum (**G**) folate level; and (**H**) homocysteine level after 12 months feeding. Data are mean ± SEM (n = 11 or 16 per group). Values were analyzed with two-way ANOVA with folate and HFF as independent factors, followed by Duncan’s post hoc test. *p* < 0.05 or different letters was considered statistically significant. (**I**) Pearson correlation between body weight and systolic blood pressure of the four groups of mice (*p* < 0.05). ns, not significant.

**Figure 2 nutrients-15-03616-f002:**
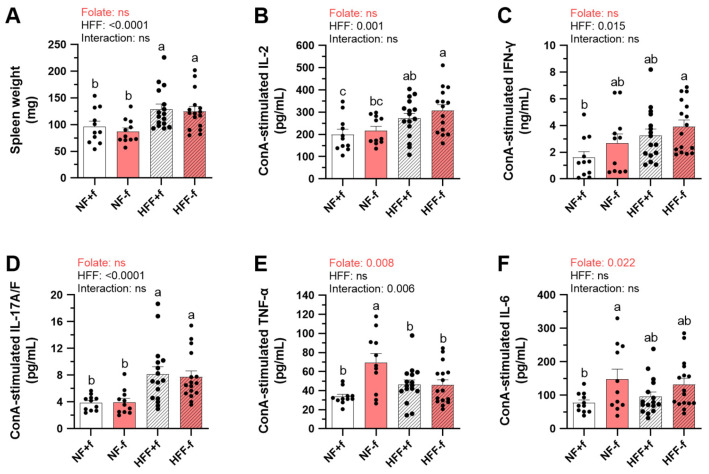
Folate deficiency increased pro-inflammatory cytokine but decreased anti-inflammatory cytokine levels produced by primary splenocytes from mice fed test diet for 12 months. (**A**) Spleen weight of NF or HFF diet-fed mice. (**B**) IL-2, (**C**) IFN-γ, (**D**) IL-17A/F, (**E**) IL-6, (**F**) TNF-α, (**G**) IL-4, and (**H**) IL-10 secreted by ConA-stimulated splenic T cells. (**I**) IL-10 secreted by LPS-stimulated splenic B cells. Data are mean ± SEM (n = 11 or 16 per group). Values were analyzed with two-way ANOVA with folate and HFF as independent factors, followed by Duncan’s post hoc test. *p* < 0.05 or different letters was considered statistically significant. ns, not significant.

**Figure 3 nutrients-15-03616-f003:**
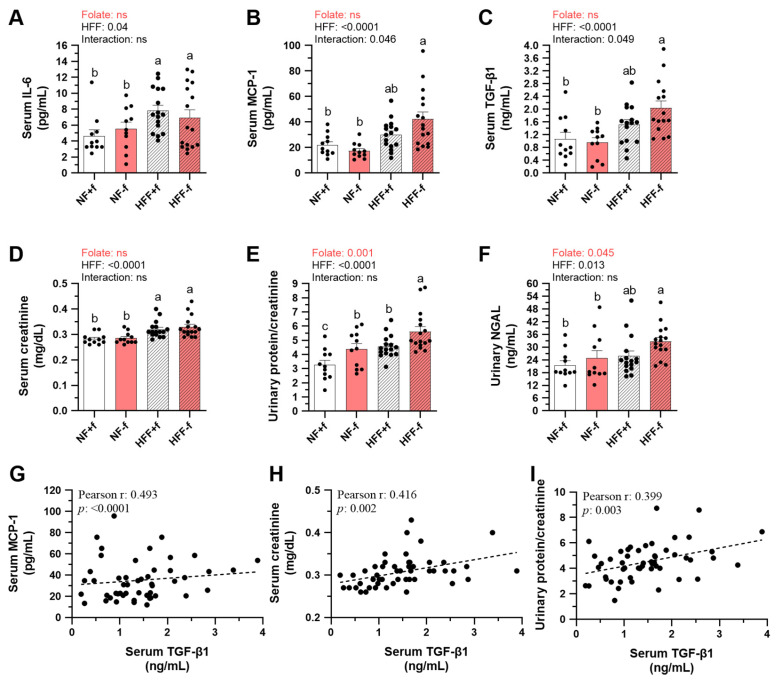
Folate deficiency increased urinary protein and NGAL excretions of mice fed test diet for 12 months. Serum (**A**) IL-6, (**B**) MCP-1, (**C**) TGF-β1, and (**D**) creatinine levels. Urinary (**E**) protein-to-creatinine ratio and (**F**) neutrophil gelatinase-associated lipocalin (NGAL) levels. Data are mean ± SEM (n = 11 or 16 per group). Values were analyzed with two-way ANOVA with folate and HFF as independent factors, followed by Duncan’s post hoc test. *p* < 0.05 or different letters was considered statistically significant. Pearson correlation between serum TGF-β1 and (**G**) MCP-1, (**H**) creatinine, or (**I**) urine protein of the four groups of mice (*p* < 0.05). ns, not significant.

**Figure 4 nutrients-15-03616-f004:**
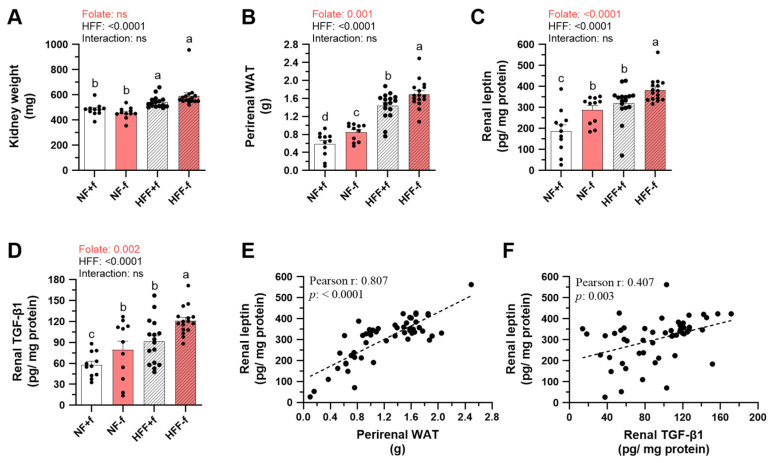
Folate deficiency elevated renal leptin and pro-fibrotic mediators of mice fed test diet for 12 months. (**A**) Kidney and (**B**) perirenal white adipose tissue (WAT) weight. Renal (**C**) leptin and (**D**) TGF-β1 levels in the kidney homogenates were determined by ELISA. Data are mean ± SEM (n =11 or 16 per group). Values were analyzed with two-way ANOVA with folate and HFF as independent factors, followed by Duncan’s post hoc test. *p* < 0.05 or different letters was considered statistically significant. Pearson correlation between renal leptin and (**E**) perirenal WAT mass or (**F**) renal TGF-β1 of the four groups of mice (*p* < 0.05). ns, not significant.

**Figure 5 nutrients-15-03616-f005:**
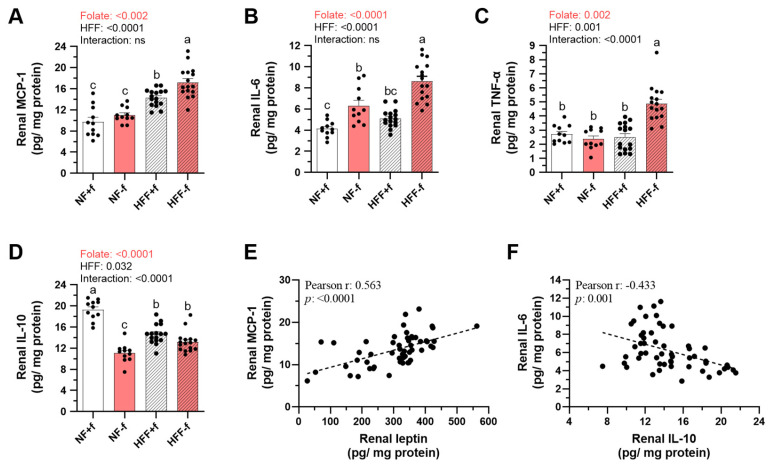
Folate deficiency increased renal pro-inflammatory cytokines of mice fed test diet for 12 months. (**A**) MCP-1, (**B**) IL-6, (**C**) TNF-α, and (**D**) IL-10 levels in the kidney homogenates were determined by ELISA. Data are mean ± SEM (n = 11 or 16 per group). Values were analyzed with two-way ANOVA with folate and HFF as independent factors, followed by Duncan’s post hoc test. *p* < 0.05 or different letters was considered statistically significant. Pearson correlation between (**E**) renal MCP-1 and leptin or (**F**) renal IL-6 and IL-10 of the four groups of mice (*p* < 0.05). ns, not significant.

**Figure 6 nutrients-15-03616-f006:**
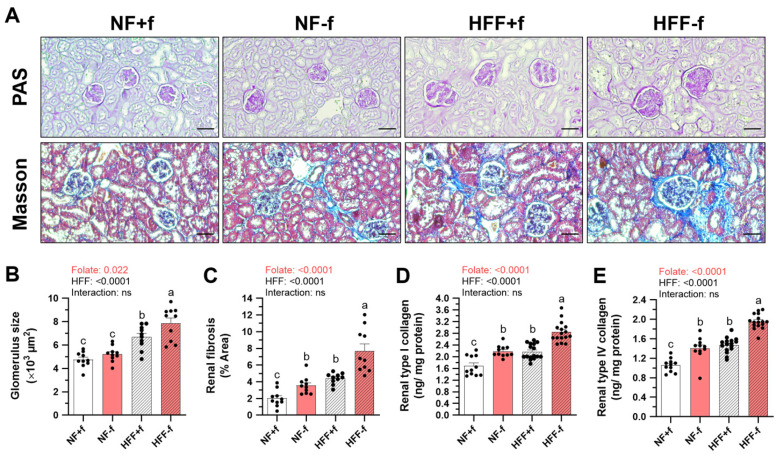
Folate deficiency exacerbated renal fibrosis and collagen depositions in mice fed test diet for 12 months. The representative images of (**A**) PAS-stained kidney (magnification, ×200; upper panel) and Masson-stained kidney (×200; lower panel). Scale bar, 50 μm. The quantification of (**B**) glomerulus size and (**C**) renal fibrosis area using ImageJ. Each value was calculated as an average of at least three sections. Data are mean ± SEM. n = 10 per group. (**D**) Type I and (**E**) type IV collagen levels in the kidney homogenates were determined by ELISA. n = 11 or 16 per group. Values were analyzed with two-way ANOVA with folate and HFF as independent factors, followed by Duncan’s post hoc test. *p* < 0.05 or different letters was considered statistically significant. ns, not significant.

**Figure 7 nutrients-15-03616-f007:**
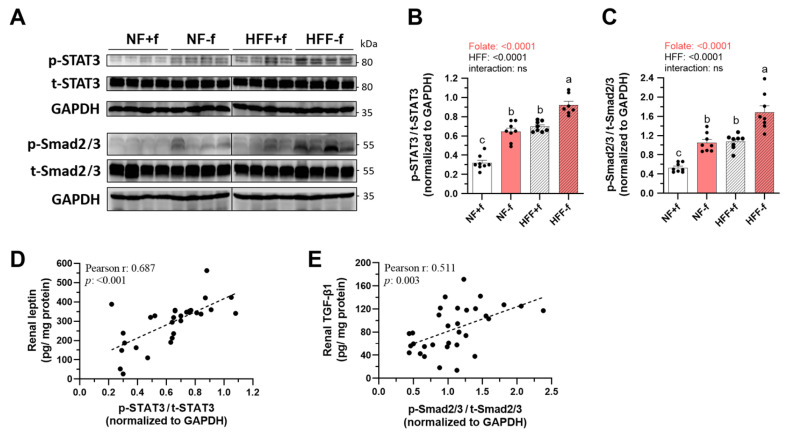
Folate deficiency enhanced renal STAT3 and Smad2/3 phosphorylation of mice fed test diet for 12 months. (**A**) Western blot and (**B**,**C**) relative quantitative analysis of renal protein expression for phosphorylated STAT3, total STAT3, phosphorylated Smad2/3, and total Smad2/3. GAPDH as loading control. Data are mean ± SEM (n = 7 or 8 per group). Values were analyzed with two-way ANOVA with folate and HFF as independent factors, followed by Duncan’s post hoc test. *p* < 0.05 or different letters was considered statistically significant. Pearson correlation between (**D**) renal leptin and p-STAT3/STAT3, (**E**) renal TGF-β1 and p-Smad2/3/Smad2/3 of the four groups mice (*p* < 0.05). ns, not significant.

**Figure 8 nutrients-15-03616-f008:**
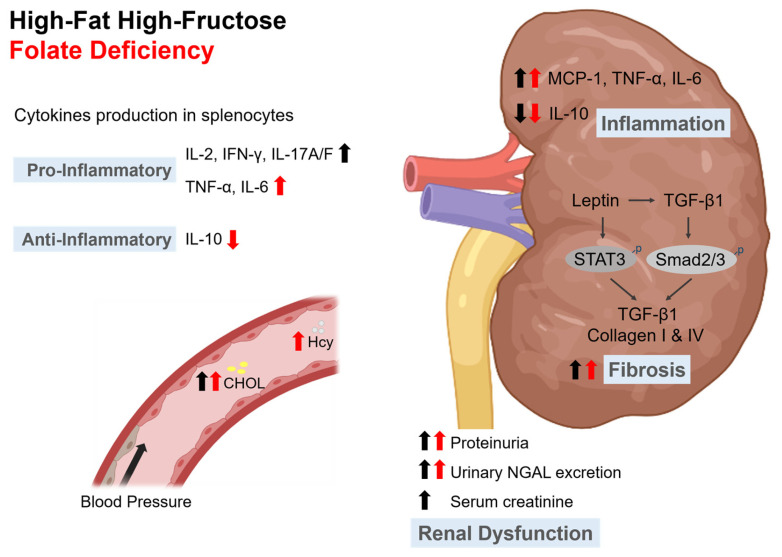
Schematic diagram of the effects of high-fat high-fructose (HFF) diet and folate deficiency (−f) on obesity-related nephropathy. Dietary −f exacerbated HFF-induced pro-inflammatory immune responses and decreased anti-inflammatory IL-10 levels. The status of those dietary habits also increased renal leptin, MCP-1, TNF-α, IL-6, and TGF-β1 levels. Both HFF and −f significantly promoted STAT3 and Smad2/3 pathways activation and subsequent TGF-β1, type I, and IV collagens production in kidneys. Thus, inadequate folate intake might exacerbate collagen deposition, renal fibrosis, and renal dysfunction in HFF-diet-fed mice. The black arrow indicates HFF effects and red arrow indicates −f effects. CHOL, cholesterol; Hcy, homocysteine. Created with BioRender.com.

## Data Availability

The original contributions presented in the study are included in the article/Appendix A; further inquiries can be directed to the corresponding author.

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
