# Peer review of "Folate Deficiency Enhanced Inflammation and Exacerbated Renal Fibrosis in High-Fat High-Fructose Diet-Fed Mice"

_nutrients, 2023, doi:10.3390/nu15163616_

Round 1
Reviewer 1 Report
Folate Deficiency Enhanced Inflammation and Exacerbated Renal Fibrosis in High-Fat High-Fructose Diet Fed Mice
Remarks to the Author:
The authors of this article investigated the impact of folate deficiency on the progression of chronic kidney disease (CKD) particularly obesity-related nephropathy in mice. To achieve this, they utilized a folate-deficient normal or high-fat high fructose (HFF) diet and found that folate deficiency increased renal inflammation and fibrosis. Additionally, the harmful effect of folate deficiency was associated with the increased phosphorylation of STAT3 and Smad2/3, cellular factors whose activation is known to induce fibrogenic responses. Overall, this study is well-designed and conducted. Also, the data mostly support the stated conclusions that significantly advance this field of research. However, this manuscript could be improved by making revisions in response to the following comments.
Specific comments:
1. The overall human relevance of the current study remains unclear. The authors at least discuss whether the levels of folate were changed in the patients with CKD (or obese patients with nephropathy) and whether the levels were correlated with disease severity.
2. In Figure 6, it is recommended that the authors include low-magnification pictures to effectively distinguish pathological differences and to make the histopathological findings more reliable.
3. The authors should evaluate whether folate deficiency affects inflammatory cell infiltration in the kidney.
4. In Figure 7A, the authors are advised to measure the fibrogenic proteins, including αSMA and collagen 1.
5. The data showed that the harmful effects of folate deficiency are leptin dependent. Does folate itself affect the production of inflammatory cytokines or the activation of signaling molecules such as STAT3 and Smad2/3? Also, treatment of folate can rescue the ConA-induced inflammation in splenocytes?
6. Does folate deficiency exacerbate another model of renal fibrosis induced by adenine, but not HFF diet feeding?
Reviewer 2 Report
Manuscript by Chan et al. is well-designed and scientifically sound.
I have a few below-mentioned suggestions
1. I would suggest changing the normal-fat diet abbreviation of groups. It is confusing to write high-fat high-fructose diet (HFF) with folate (HFF-f1, and HFF without folate (HFF-f0). Authors can write HFF+f and HFF-f instead.
2. I would suggest changing bar graphs to dot bar graphs and having all the statistics significant (stars) on the bar graphs clearly indicating group comparisons.
3. Cite or briefly write PAS staining protocol method section of 2.7.
4. Western blot method needs more details section 2.9.
5. Figure 8- I would suggest changing the balance to lower in the IL10 side and high in TNFa and IL-6.
Round 2
Reviewer 1 Report
All issues are well addressed for publication.